# Quantum Deep Equilibrium Models

**Philipp Schleich**[*]
Department of Computer Science
University of Toronto
Vector Institute
philipps@cs.toronto.edu

**Marta Skreta**[*]
Department of Computer Science
University of Toronto
Vector Institute
martaskreta@cs.toronto.edu

**Lasse B. Kristensen**
Department of Computer Science
University of Copenhagen

**Rodrigo A. Vargas-Hernández**
Department of Chemistry
& Chemical Biology
McMaster University, ON

**Alán Aspuru-Guzik**
Department of Computer Science
Department of Chemistry
University of Toronto
Vector Institute

## Abstract

The feasibility of variational quantum algorithms, the most popular correspondent of neural networks on noisy, near-term quantum hardware, is highly impacted by the circuit depth of the involved parametrized quantum circuits (PQCs). Higher depth increases expressivity, but also results in a detrimental accumulation of errors. Furthermore, the number of parameters involved in the PQC significantly influences the performance through the necessary number of measurements to evaluate gradients, which scales linearly with the number of parameters. Motivated by this, we look at deep equilibrium models (DEQs), which mimic an infinite-depth, weight-tied network using a fraction of the memory by employing a root solver to find the fixed points of the network. In this work, we present Quantum Deep Equilibrium Models (QDEQs): a training paradigm that learns parameters of a quantum machine learning model given by a PQC using DEQs. To our knowledge, no work has yet explored the application of DEQs to QML models. We apply QDEQs to find the parameters of a quantum circuit in two settings: the first involves classifying MNIST-4 digits with 4 qubits; the second extends it to 10 classes of MNIST, FashionMNIST and CIFAR. We find that QDEQ is not only competitive with comparable existing baseline models, but also achieves higher performance than a network with 5 times more layers. This demonstrates that the QDEQ paradigm can be used to develop significantly more shallow quantum circuits for a given task, something which is essential for the utility of near-term quantum computers. Our code is available at https://github.com/martaskrt/qdeq.

---

[*]Equal contribution. Author order was sampled from a quantum computer.

38th Conference on Neural Information Processing Systems (NeurIPS 2024).

# 1 Introduction

Quantum computing holds a lot of theoretical promise for transforming the computational landscape of a variety of applications. Machine learning is one of these applications (Biamonte et al., 2017), enabled by the capability of quantum computing to handle large amounts of data with significant quantum speedups and by the fact that dimensionality of the output is typically much smaller than the one of the input and during processing, a requirement to recover linear-algebra relevant speedups (Aaronson, 2015). Although there have been tremendous advances in hardware development in recent years, the time of full-stack, error corrected quantum computers which would enable such linear-algebra speedups is still not yet within tangible reach. Therefore, advances in near-term quantum algorithms for noisy devices with short coherence times remain one of the most promising avenues toward effectively harnessing quantum computing for real-world applications.

The arguably most studied class of algorithms in this line of research is the class of variational quantum algorithms (VQAs). While they originated as a means of physics-inspired learning without data (Peruzzo et al., 2014), they provide the most natural way to extend classical learning models to a hybrid quantum-classical setting. In VQAs, a parametrized quantum circuit (PQC) together with measurement of suitable observables on the quantum state after the circuit gives rise to a function class. Through the choice of a quantum circuit, observables, and loss function, this setup allows different problems to be tackled, with classical optimization of the loss function as the training step. Within this work, we use a VQA for classification tasks. Challenges within VQA training continue to be limited circuit depth, which reduces the the expressibility and trainability of the circuit, and disadvantageous loss landscapes, influenced by induced entanglement and the specific loss function (McClean et al., 2018; Fontana et al., 2024; Ragone et al., 2024). Further, while the parameter-shift rule allows us to evaluate gradients exactly using a simple finite-difference formula, there is a significant measurement overhead to determine the gradients. Thus, striving for training methods that enable the use of shallower circuits with less independent parameters at similar performance is paramount.

The main contribution of our work is the proposition of a class of quantum deep equilibrium models (QDEQ). To our knowledge, this is the first application of deep equilibrium models (DEQ) as first introduced in Bai et al. (2019) in the context of quantum computing; so far, previous work has explored implicit differentiation techniques for PQCs (Ahmed et al., 2022). Implicit and adjoint methods such as DEQs in machine learning stand out by their memory efficiency compared to their explicit counterparts. While this is still true for QDEQ, we see the main benefit in the reasons outlined in the paragraph above, i.e., the ability to use shallower circuits with fewer independent parameters to tackle a given problem. We will give some theoretical intuition why DEQ can be expected to perform well on a specific family of quantum model functions in Section 2.3, design a set of numerical experiments in Section 3 and obtain numerical results that confirm this hypothesis in Section 4.

# 2 Background and related works

## 2.1 Deep Equilibrium Models

Based on the fact that a neural network comprised of $L$ layers is equivalent to an input-injected, weight-tied network of the same depth, Bai et al. (2019) introduced the concept of Deep Equilibrium Models. Assuming that the respective layer $f_\theta(\cdot)$ has a fixed point (Winston and Kolter, 2020), instead of explicitly repeating the weight-tied layer $L$ times, we solve

$$f_\theta(\mathbf{z}^\star; \mathbf{x}) - \mathbf{z}^\star = g_\theta(\mathbf{z}^\star; \mathbf{x}) = 0 \tag{1}$$

using a black-box root finder such as Newton's or Broyden's method. The latter is comprised by Newton iterations where the Jacobian is approximated by a finite difference formula (Broyden, 1965). Finding this fixed point corresponds to evaluating an infinitely deep network, in the sense that $\lim_{L\to\infty} \mathbf{z}^{(L)} = \lim_{L\to\infty} \underbrace{(f_\theta \circ \cdots \circ f_\theta)}_{L \text{ times}}(\mathbf{z}^{(0)}) = \mathbf{z}^\star$.

When training this model with respect to a loss function $\ell$, even though we use a root-finder to evaluate the model, we still need to be able to differentiate the procedure for training. Instead of explicitly differentiating through the $L$ repetitions of the same layer (we call this later the DIRECT solver), Bai et al. (2019) make use of the implicit function theorem (Krantz and Parks, 2013) to

perform implicit differentiation in the following sense. During backpropagation, the gradient of the loss function can be expressed as:

$$\frac{\partial \ell}{\partial \theta} = -\frac{\partial \ell}{\partial \mathbf{z}^\star}\left(J_{g_\theta}^{-1}\mid_{\mathbf{z}^\star}\right)\frac{\partial f_\theta(\mathbf{z}^\star; \mathbf{x})}{\partial \theta}, \tag{2}$$

with $-\left(J_{g_\theta}^{-1}\mid_{\mathbf{z}^\star}\right) = (I - J_{f_\theta}\mid_{\mathbf{z}^\star})^{-1}$. While the last factor in Eq. (2) can be computed by standard automatic differentiation through the model function, determining $-\frac{\partial \ell}{\partial \mathbf{z}^\star}\left(J_{g_\theta}^{-1}\mid_{\mathbf{z}^\star}\right)$ either requires assembling the full Jacobian and inverting it (which is quite expensive) or solving an additional root-finding problem during the backward pass in form of solving for $\mathbf{q}$ in the equation

$$\left(J_{g_\theta}^\mathrm{T}\mid_{\mathbf{z}^\star}\right)\mathbf{q}^\mathrm{T} + \left(\frac{\partial \ell}{\partial \mathbf{z}^\star}\right)^\mathrm{T} = 0, \tag{3}$$

with gradients computed efficiently using the vector-Jacobian product.

The overall procedure of a DEQ model thus is to minimize a loss function $\ell$ given input $\mathbf{x}$, target $y$, and a hypothesis class in the following way: In the backward pass, we use the gradient expression in Eq. (2) and determine the first factor using a black-box root finder on Eq. (3). Using these gradients, the model is trained. The model is then evaluated on new input by using a root-finder to solve Eq. (1). A brief overview of this procedure is in Fig. 1.

It was found that pre-training the DEQ model using a shallow (2-layer), weight-tied DIRECT approach – that is, explicit application and training of $f \circ f$ – can considerably aid the overall convergence behaviour (Bai et al., 2019, 2020). This pretraining step is a cheap way to give the model reasonable weights as a starting point (called a warm-up), but the model is too shallow to achieve the best performance. In this work, we will utilize this warm-up strategy, then use implicit differentiation to optimize the loss and obtain better accuracy. We call this the IMPLICIT+WARMUP solver, and refer to training using only implicit differentiation as the IMPLICIT solver.

An important step to ensure stability of the root-finding algorithm is regularizing the Jacobian, as introduced in Bai et al. (2021). Further advances include a convenient inclusion to `pytorch` through Geng and Kolter (2023) and certifiable (Li et al., 2022) and robust (Chu et al., 2023) DEQ models as well as DEQ for diffusion models (Pokle et al., 2022).

## 2.2 Quantum Models and Variational Algorithms for Classification

Variational quantum algorithms rely on a PQC over $Q$ qubits, which span the finite-dimensional Hilbert space $\mathbb{C}^{2^Q}$. Then, $\{|i\rangle\}_{i=0}^{2^Q-1} \sim \{|i_1\rangle \otimes |i_2\rangle \otimes \cdots |i_n\rangle\}_{i_1,\ldots,i_n \in \{0,1\}}$ is an orthonormal basis for this space and $\langle i|$ is the conjugate transpose to a vector $|i\rangle$. We denote the PQC, parametrized by $p$ parameters, by the $Q$-qubit unitary operation $U(\theta) \in \mathbb{C}^{2^Q \times 2^Q}$. Then, one can define an objective $\langle \phi| U^\dagger(\theta) M U(\theta) |\phi\rangle$ that is to be variationally minimized, induced by a Hermitian matrix $M \in \mathbb{C}^{2^Q \times 2^Q}$ and an initial state $|\phi\rangle \in \mathbb{C}^{2^Q}$ (Bharti et al., 2022; Cerezo et al., 2021). Quantum machine learning models have already been the target of a fair amount of study, yielding insights into several properties, including generalization bounds (Caro et al., 2022), their training in general (Beer et al., 2020), and interpretability (Pira and Ferrie, 2024).

So far, we have not discussed how this framework is able to incorporate data into the training; here, this role will be filled by the initial state $|\phi\rangle$. Given classical data $\mathbf{x} \in \mathbb{R}^n$, there exists a unitary data encoding circuit $S_\mathbf{x}$ that maps the data in some fashion onto a quantum computer and stores it in a state $|\mathbf{x}\rangle$; there is a variety of techniques to do so, which e.g. stores the data in the amplitudes (coefficients) of the state, applies rotations parametrized by the data, etc. (Schuld et al., 2020; Schuld, 2021). Then, we can define a class of quantum model functions

**Definition 1** (Family of Quantum Model Functions). *Let $U(\theta)$ be a unitary circuit over $Q$ qubits parametrized by $\theta \in \mathbb{R}^p$, $\mathbf{x} \in \mathbb{R}^n$ features and $S_\mathbf{x} : |0\rangle \mapsto |\mathbf{x}\rangle$ a data encoding. Then, we define a quantum model function as*

$$f_\theta^M(\mathbf{x}) = \langle \mathbf{x}| U^\dagger(\theta) M U(\theta) |\mathbf{x}\rangle, \tag{4}$$

*for a Hermitian observable $M$. In order to distinguish $K$ labels for a classification problem, we need an ensemble of size $K$. To that end, we define an map $R' : \mathbb{C}^{2^Q \times 2^Q} \to \mathbb{R}^K$ that describes measurement of expectation values with respect to an ensemble of $K$ observables $\{M\}$ and storage of the respective outcomes in a $K$-dimensional vector. Typically, we string this together with an*

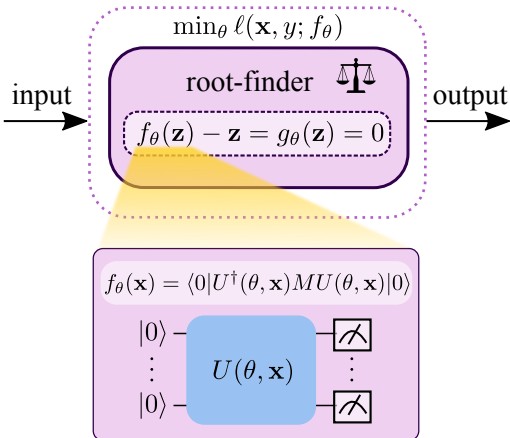

Figure 1: Instance of a Deep Equilibrium Model using a quantum model family. A black-box root-finding method is used to determine the model function's equilibrium state.

*upsampling isometry $\mathcal{I}_u$ so that $R = \mathcal{I}_u \circ R'$ maps to $\mathbb{R}^n$, which allows repeated calls to reach multiple layers. This gives rise to a family of model functions $f_\theta^{\{M\}}$,*

$$f_\theta^{\{M\}}(\mathbf{x}) = R\left(U(\theta)\,|\mathbf{x}\rangle\langle\mathbf{x}|\,U^\dagger(\theta)\{M\}\right). \tag{5}$$

We note that the model given in Definition 1 combined with activation functions and a bias vector was shown to be universal in Hou et al. (2023), whilst Schuld et al. (2020) provides evidence that for classification problems, as we will consider below, the architecture in Definition 1 is sufficient.

Training of such a model can proceed equivalently to the training of variational models (Bharti et al., 2022; Cerezo et al., 2021), where one defines loss functions based upon the outputs of the quantum model family $f_\theta^{\{M\}}$. While backpropagation is possible in a QML framework, it is rather uncommonly done. As was shown in Abbas et al. (2021), achieving the computational efficiency of backpropagation in classical learning models also in quantum models turns out to be very challenging and resource demanding. Most approaches to gradient evaluation thus rely on the so-called parameter-shift rule (Schuld and Killoran, 2019), where gradients can be computed exactly using an approach similar to central finite differences. The requirement for this to work is that the generator $g$ of each unitary gate $V(\theta) = \mathrm{e}^{-\mathrm{i}\frac{\theta}{2}g}$ has eigenvalues $\pm r$. Then, $\partial_\theta f_\theta = r[f_{\theta+s} - f_{\theta-s}]$ for $s = \frac{\pi}{4r}$, where we note that for $p$ parameters $\theta \in \mathbb{R}^p$, this gradient computation needs to be carried out for each parameter individually. Generalizations beyond two symmetric eigenvalues have been done and are in further development (Kottmann et al., 2021; Hubregtsen et al., 2022; Wierichs et al., 2022; Anselmetti et al., 2021; Kyriienko and Elfving, 2021). The gradient computation here needs to measure two expectation values per parameter up to sampling accuracy via the parameter-shift rule. For considerations on an implicit model, we expect a similar measurement overhead for the extraction of the Jacobian through a finite difference approximation in Broyden's method. When parameter shift rules are applicable, the finite differences in Broyden's method can be made exact up to sampling accuracy as well.

## 2.3 A Quantum Deep Equilibrium Model

We next introduce Quantum Deep Equilibrium Models, based on the notion of DEQs with the quantum model families from Definition 1. Fig. 1 depicts the overall process: In the course of minimizing a loss function, we consider the solution of a root finder of the quantum model family as the hidden layer.

While QDEQ is not restricted to classification tasks, this is what we will use as the context in our work. To perform classification, we use measurement ensembles of dimension $K$, as necessary for $K$ different classes. Additionally, since the quantum basis scales exponentially in the number of qubits, we will often keep $K$ significantly below this exponential threshold. Note that one option for

estimating these observables would be to use shadow tomography to obtain a set of classical shadows that can be used for observable estimation (Huang et al., 2020).

Given the considerations above, one choice for a measurement ensemble is a set of basis state projections $\{|k\rangle\langle k|\}_{k=1}^{K}$. This choice allows one to extract more data than the number of qubits; in fact, up to as many as the number of basis states, however this comes with a similar cost as state tomography, thus ideally we choose $K$ well below $2^Q$. Another choice of ensemble, which fits the framework of DEQs with input-injection better are Pauli measurements. This is because every Pauli measurement result falls within $[-1; 1]$ per qubit and thus directly gives a notion of perturbing the injected input towards smaller and larger. Basis state projections yield strictly positive measurement results, allowing only positive perturbations of the input injection, and tend to be smaller in magnitude as they represent probabilities. In the sense of Lloyd et al. (2020), the former corresponds to the fidelity classifier and the latter has similarities with what they call Helstrøm classifier.

We will use quantum model functions for classification into $K$ classes and build our arguments on the following observations, which we discuss in more detail in the appendix:

*(i)* Such classes of models admit fixed-points, a necessary condition for DEQ frameworks to be successful. See Observation 2. The argument is based on the property how quantum models encode and extract information through measurements.

*(ii)* Weight-tied QDEQ with input injection is equivalent to a sequence of independently parametrized layers. See Theorem 3; extending Theorem 3 in Bai et al. (2019).

In combination, this gives evidence that the quantum model in Definition 1 with a measurement ensemble of length $K$ can be successfully applied to classification using DEQ strategies. We are able to confirm this through our numerical experiments in the next section.

The layer depicted in Fig. 1 also contains measurement operations. As such, QDEQ does not correspond to evaluating an infinitely deep unitary quantum circuit; a layer here means quantum model including encoding and measurement. Direct application of the ideas of this paper in the context of infinitely deep circuits is hampered by both the conceptual difficulty of reasoning about fixed points for unitary isometries and the exponentially scaling effort required to extract and compare descriptions of quantum states, beyond classical shadows Huang et al. (2020).

## 3 Experiments

### 3.1 Datasets

We consider three datasets in this study. First, we consider MNIST-4, which consists of 4 classes of MNIST digits (0, 3, 6, 9) (Deng, 2012). We then extend our model to all 10 classes of MNIST (MNIST-10), as well as FashionMNIST (FashionMNIST-10) (Xiao et al., 2017). Finally, we tested our setup on natural images with CIFAR-10 Krizhevsky et al. (2009). For all datasets, we used default train/test splits[2] and randomly split the training set into 80% train, 20% validation.

### 3.2 Architecture

We replicate the circuit from Wang et al. (2022a) as our architecture, shown in Fig. 2. The circuit consists of four qubits. We downsample the MNIST-4 images from 28x28 to 4x4 using average pooling. For encoding, we look at the options of an angle encoding through rotation gates, as in Wang et al. (2022a), or an amplitude encoding, as implemented in `torchquantum` (Wang et al., 2022a). We pass the information through parameterized quantum gates and, finally, we measure the qubits to get a readout value from the circuit. Thus far the definition of our quantum model. Then, we transform this readout value using a small classifier head, which consists of a linear layer, then pass the result into a cross-entropy loss. As in Bai et al. (2019), we also incorporate variational dropout in the classifier head. For simulations with 10 classes, we extend this circuit by repeating the four-qubit circuit with a stride of 2 qubits in a stair-case manner, as shown in Fig. 3.

The measurement ensemble we choose for our simulations is also in alignment with Wang et al. (2022a), namely a Pauli-$Z$ matrix per qubit, i.e. $\{M\} = \{Z \otimes I_{Q-1}, I \otimes Z \otimes I_{Q-2}, \ldots, I_{Q-1} \otimes Z\}$.

---

[2] https://pytorch.org/vision/stable/datasets.html

Note that the baselines Dilip et al. (2022); Shen et al. (2024) use linear maps based on measurements of basis state projections. This introduces another layer that needs training and, most importantly, there is a risk that more general observables induce barren plateaus due to non-local support (Fontana et al., 2024; Ragone et al., 2024). Furthermore, their measurement will be increasingly costly as overlap measurement through Hadamard tests is more involved that single-qubit observables.

When it comes to data-encoding, we tested both angle and amplitude encodings for the 4 and 10 qubit experiments and generally found that the performance of the amplitude encoding is superior, both for DIRECT and IMPLICIT scenarios. Thus, we present the results for the datasets with 10 classes in the subsequent section only for amplitude encodings and show both for MNIST-4.

### 3.3 Models and baselines

For each dataset, we test our setup on six model variations. The first is our IMPLICIT framework. The next four are DIRECT solvers, which are weight-tied networks consisting of $L$ repeated layers of the quantum model, where $L \in \{1, 2, 5, 10\}$. We explicitly differentiate through these layers to compare with implicit differentiation and understand whether increasing network depth results in better performance. The intuition is that if DIRECT with $L$ repetitions shows a positive trend in performance on a task with increasing $L$, applying the IMPLICIT solver should be expected to be beneficial. Finally, it was found in (Bai et al., 2019, 2020) that pre-training the DEQ model using a shallow (2-layer), weight-tied DIRECT approach can considerably aid the overall convergence behaviour. This pretraining step is a cheap way to give the model reasonable weights as a starting point (called a warm-up), but the model is too shallow to achieve the best performance. We then use implicit differentiation to optimize the loss and obtain better accuracy. We call this the IMPLICIT+WARMUP solver.

In the IMPLICIT approaches, we pass the measurements from the quantum circuit into the root finder to get $\mathbf{z}^\star$, which we then pass to the classifier head and update the parameters using implicit differentiation. We also inject the original image into every iteration of the Broyden solver, in alignment with the arguments in Bai et al. (2019) and our universality theorem in Theorem 3, as input-injected weight-tied models can be phrased equivalently to models that are not weight-tied. We train the implicit models using a Broyden solver for at most 10 steps. For optimization, we use Adam (Kingma and Ba, 2014) and cross-entropy loss. We trained each model for 100 *total* epochs (i.e. if we first pre-trained using $x$ warm-up epochs, we then trained using the implicit framework for $100 - x$ epochs) (for CIFAR-10, we only trained for 25 total epochs since we found it to converge faster). We selected hyperparameters using the validation set; see Appendix E.

Finally, for each dataset, we report the results of baselines from literature. In relation to choice of baseline, the field of quantum computational image classification is rich, with several architectures having been developed for this specific purpose, often inspired by classical convolutional neural networks (Liu et al., 2021; Henderson et al., 2020). While such models can display impressive performance on the tasks investigated here, they often employ significant classical and quantum resources in the form of classical neural networks or multiple different quantum circuit evaluations. More importantly, they represent highly specialized architectures. Since the goal of the paper is to investigate the applicability of DEQ training on general quantum models, the models chosen for this study were not highly specialized for image classification. The merit of DEQ should therefore be evaluated by comparison to results for similarly general models rather than the state-of-the-art accuracies of larger specialized models of 97% on MNIST (Henderson et al., 2020) and 83% on Fashion-MNIST (Jing et al., 2022) .

**Baseline for MNIST-4** For MNIST-4, we can directly compare our results to Wang et al. (2022b) as we chose the same circuit. The difference compared to a DIRECT realization of our model with one layer is in the ultimate classification layer. The architecture can be found in Fig. 2, for more details we refer to Section 3.2.

An alternative baseline is available in West et al. (2024). For the MNIST-4 case, where they use different classes (digits 0-3) than we did, they use five qubits, compared to four in our case. Data is encoded in amplitudes. The variational circuit they use for training is built from general unitaries and they mention that they employ 20 layers.

**Baseline for MNIST-10** As the topic under evaluation is the concept of QDEQ training, the most applicable comparison is with the same circuit trained using DIRECT solver. However, we also include results by Alam et al. (2021) using a model of similar complexity. Their approach similarly

uses 10 qubits, as well as a similar measurement ensemble. However, beyond the training strategy, the two approaches differ in encoding strategy. While we perform a simple image scaling to a $N = 10 \times 10 = 100$ pixel image and then do $O(N)$-depth amplitude encoding in $O(\log(N))$ qubits, they do simple depth-1 angle encoding in $O(N)$ qubits, facilitated by using more complex image compression by either PCA or a trained variational autoencoder. Since they demonstrate that compression scheme has a large impact on performance (see Table 2), this makes direct comparison somewhat difficult. As for MNIST-4, West et al. (2024) is another baseline here; for MNIST with all ten classes, they used 200 layers in training, making the circuit significantly more complex than ours.

**Baseline for FashionMNIST-10** For FashionMNIST, we look at the classifier circuit from Dilip et al. (2022) as a baseline because their smallest setup comes closest to our architecture, consisting of circuits of a comparable scale and number of qubits. However, the comparison is again difficult, since the results vary considerably depending on the quality of input state encoding they choose. For their encoding, quantum circuits are trained to generate approximations of amplitude-encoding-like states, called an FRQI states (Le et al., 2011), corresponding to each image. Nevertheless, high-quality FRQI states may be comparable to the amplitude encoded states used in our work. Similar work in Shen et al. (2024) additionally provide results either using amplitude encoding or using approximate FQRI and similar classifier heads to our setup. While these results use circuits with a larger number of trainable parameters (thus, likely higher expressivity), they are included for context as well. Finally, West et al. (2024) is again another baseline with equivalent setup to MNIST with all ten classes.

## 4 Results

In this section, we report on the result of our model on the datasets MNIST-4, MNIST-10, FashionMNIST-10, and CIFAR-10. As mentioned, all results were generated using the `torchquantum` framework (Wang et al., 2022a). Note that in the sections below, while the memory observed is in terms of classical memory that has been used for a simulation of the QML model, this is a rough measure for the number of parameters in the optimization and thus also quantifies the measurement overhead for QML models. Thus, we can expect this quantity to be a reasonable estimate, comparing different approaches relative to one another, for the necessary resources for evaluation on quantum hardware.

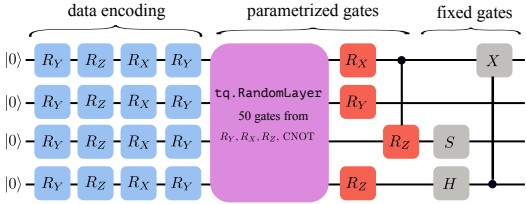

(a) 4x4 YZXY Angle encoding.

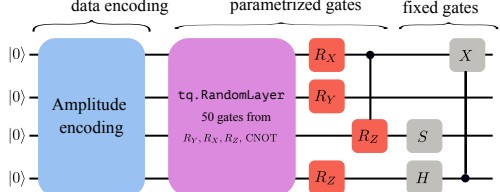

(b) Amplitude encoding instead of angle encoding from (a), using the implementation in `torchquantum`.

Figure 2: Circuit used for classification with up to 4 classes, following Wang et al. (2022a). Blue circuits correspond to input, purple and red shades to parametrized and trainable gates and grey to fixed gates. The RandomLayer has on average 12.5 = 50/4 two-qubit gates (CNOTs).

### 4.1 MNIST-4

In Table 1, we show the performance of our proposed QDEQ framework on the MNIST-4 test set. We observed that for angle encoding, deeper DIRECT models seems to perform better than shallow ones, as expected. However, the IMPLICIT models outperforms all of the DIRECT models, including the baseline. In contrast, amplitude encoding has the shallow DIRECT models perform best, and better performance in general. This may indicate both that deeper DIRECT models can suffer trainability issues compared to shallower DIRECT models, and that QDEQ does not always provide a benefit. However, as we will for MNIST-10 see below, these two effects are not linked, and need not co-occur. In addition to this, we provide a result from West et al. (2024) with a circuit of 20 layers, thus expectedly much deeper and more variational parameters than our circuit. They achieve an accuracy

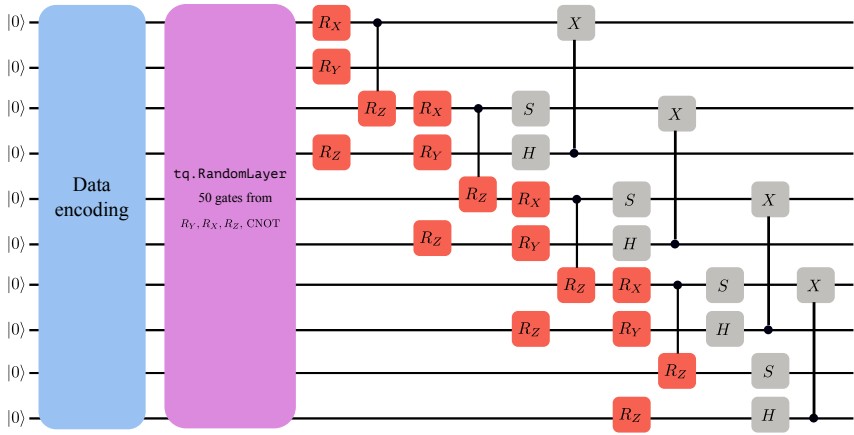

Figure 3: Extension of the circuit presented in Wang et al. (2022b) to a scenario of 10 qubits. Similarly to the tensor-network inspired circuits in Dilip et al. (2022), we repeat the four-qubit stencils in a staircase manner.

of approximately 90-92%. This supports our hypothesis that the QDEQ model is able to achieve high accuracies (up to 93.4%) at lower cost.

Table 1: Model performance on MNIST-4. Memory refers to the maximum GPU memory occupied by tensors. Runtime was calculated over 100 epochs on a NVIDIA RTX 2070 GPU. We present results both for an amplitude and angle encoding as shown in Fig. 2.

| Model | Test accuracy (%) | Memory (MB) | Runtime (sec) | Residual |
|---|---|---|---|---|
| Amplitude Encoding | | | | |
| QML Circuit [Wang et al. (2022b)] / `torchquantum` example[3] | 85.3 | 24.49 | 4257.17 | – |
| IMPLICIT solver [**ours**] | 92.9 | 22.0 | 3206.43 | 2.395e-4 |
| IMPLICIT+WARMUP solver [**ours**] | 93.4 | 22.0 | 5276.61 | 6.982e-4 |
| DIRECT solver - 10 layers [**ours**] | 91.8 | 33.3 | 7820.00 | 9.877e-4 |
| DIRECT solver - 5 layers [**ours**] | 91.0 | 26.1 | 6013.00 | 0.189 |
| DIRECT solver - 2 layers [**ours**] | 92.1 | 21.9 | 3126.66 | 1.003 |
| DIRECT solver - 1 layers [**ours**] | 93.5 | 20.4 | 2566.00 | 2.867 |
| Angle Encoding | | | | |
| QML Circuit [Wang et al. (2022b)] | 77.3 | 24.49 | 4762.98 | – |
| IMPLICIT solver [**ours**] | 85.8 | 21.23 | 3206.43 | 5.115e-4 |
| IMPLICIT+WARMUP solver [**ours**] | 86.7 | 22.55 | 2306.72 | 1.037e-3 |
| DIRECT solver - 10 layers [**ours**] | 84.8 | 39.38 | 3581.83 | 1.047 |
| DIRECT solver - 5 layers [**ours**] | 85.3 | 28.86 | 1926.27 | 1.818 |
| DIRECT solver - 2 layers [**ours**] | 82.7 | 22.55 | 1012.66 | 3.625 |
| DIRECT solver - 1 layers [**ours**] | 83.9 | 20.45 | 704.38 | 4.135 |

## 4.2 MNIST-10 and FashionMNIST-10

We then extend our circuit from 4 classes to 10 and show the performance on MNIST-10 (Table 2) and FashionMNIST-10 (Table 3). In both cases, the IMPLICIT framework performs better than DIRECT circuits that are 3 or more times larger in terms of memory required.

For MNIST-10, we observe that the deeper models using DIRECT training perform comparably or slightly worse than the shallowest DIRECT models. Similarly to for MNIST-4, this could indicate suboptimal training for deeper circuits. In contrast, the IMPLICIT models seem to circumvent

this problem, and slightly outperform both the DIRECT models and the baseline using PCA-based encoding. The VAE-based baseline on the other hand shows significantly higher performance, showing that such approaches to data compression (and their combination with QDEQ) may be a worthy target for further study. The results in West et al. (2024) for the all MNIST classes reach up to 79.3% (Figure 2 in their paper); while this is more accurate than ours (up to 73.68%), this can be ascribed to the vastly larger model.

For FashionMNIST-10, we observe performance differences between DIRECT and IMPLICIT models very similar to the MNIST-10 case. Furthermore, performance is competitive with the baseline results of Dilip et al. (2022), possibly reflecting a larger similarity between the encoding methods (amplitude encoding and approximate FRQI, respectively). The case for West et al. (2024) is the same as for MNIST above; with an accuracy of 74.5% (Figure 2 in their paper), they surpass our best test accuracy of 72.11%, while again, their model is far larger. Similarly, we ascribe the performance delta with Shen et al. (2024) to a similar difference in the number of trainable parameters.

Table 2: Model performance on MNIST-10. Memory refers to the maximum GPU memory occupied by tensors. Runtime was calculated over 100 epochs on a NVIDIA RTX 2070 GPU.

| Model | Test accuracy (%) | Memory (MB) | Runtime (sec) | Residual |
|---|---|---|---|---|
| QML Circuit (VAE) [Alam et al. (2021)] | 89.80 | - | - | - |
| QML Circuit (PCA) [Alam et al. (2021)] | 71.75 | - | - | - |
| IMPLICIT solver [**ours**] | 73.68 | 273.1 | 7301.74 | 5.598e-3 |
| IMPLICIT+WARMUP solver [**ours**] | 73.33 | 273.1 | 7017.38 | 1.583e-3 |
| DIRECT solver - 10 layers [**ours**] | 71.18 | 1233.3 | 2856.94 | 1.905e-4 |
| DIRECT solver - 5 layers [**ours**] | 72.46 | 628.8 | 1737.82 | 2.093e-3 |
| DIRECT solver - 2 layers [**ours**] | 72.07 | 266.1 | 4304.54 | 0.596 |
| DIRECT solver - 1 layers [**ours**] | 72.78 | 145.2 | 3343.40 | 4.026 |

Table 3: Model performance on FashionMNIST-10. Memory refers to the maximum GPU memory occupied by tensors. Runtime was calculated over 100 epochs on a NVIDIA RTX 2070 GPU. For Dilip et al. (2022), performance depends on encoding quality, and thus a range is given, while two comparable baselines are included for Shen et al. (2024) (see Section 3.3 for details).

| Model | Test accuracy (%) | Memory (MB) | Runtime (sec) | Residual |
|---|---|---|---|---|
| QML Circuit [Dilip et al. (2022)] | 63-75 | - | - | - |
| QML Circuit [Shen et al. (2024)] | 77-80 | - | - | - |
| IMPLICIT solver [**ours**] | 72.11 | 273.1 | 6424.67 | 6.892e-3 |
| IMPLICIT+WARMUP solver [**ours**] | 71.17 | 273.1 | 7240.35 | 1.934e-3 |
| DIRECT solver - 10 layers [**ours**] | 71.83 | 1233.3 | 11651.96 | 1.41e-5 |
| DIRECT solver - 5 layers [**ours**] | 70.87 | 628.8 | 6894.50 | 4.803e-3 |
| DIRECT solver - 2 layers [**ours**] | 70.81 | 266.1 | 4135.23 | 0.210 |
| DIRECT solver - 1 layers [**ours**] | 71.05 | 145.2 | 3382.09 | 1.431 |

## 4.3 CIFAR-10

Finally, we applied our method to CIFAR-10, a dataset of natural images; results can be found in Table 4. We find that in general, near-term quantum ML models that are amenable to numerical experiments do not perform well on CIFAR-10, as observed in recent prior work (Baek et al., 2023; Monbroussou et al., 2024). While their performance is higher than ours, we note that our models are not directly comparable. We chose not to boost model performance by adding classical NN components to focus on studying the quantum model. Still, we find that our QDEQ framework (IMPLICIT+WARMUP ) has higher accuracy on the test set than the direct solver baseline. This motivates the utility of our method on more realistic datasets.

Table 4: Model performance on CIFAR-10.

| Model | Test accuracy (%) |
|---|---|
| IMPLICIT  solver [**ours**] | 24.38 |
| IMPLICIT+WARMUP  solver [**ours**] | **25.45** |
| DIRECT  solver - 10 layers [**ours**] | 23.71 |
| DIRECT  solver - 5 layers [**ours**] | 24.19 |
| DIRECT  solver - 2 layers [**ours**] | 24.90 |
| DIRECT  solver - 1 layers [**ours**] | 24.70 |

## 5   Conclusion and open problems

In this work, we introduce a novel approach to training QML models under the framework of deep equilibrium models. We theoretically demonstrate that under certain conditions, QML models for classification tasks are amenable to DEQ-based optimization. This finding aligns with our numerical observations, suggesting that DEQs can significantly reduce circuit complexity while maintaining or improving performance compared to explicit training methods; this benefit is expected to hold in particular compared to using deeper parametrized circuits in quantum neural networks to achieve similar expressibility to the QDEQ model. This is particularly advantageous for near-term quantum algorithms, where minimizing circuit depth is crucial.

Our numerical experiments indicate that, for the case of quantum model families with theoretical intuition about their fixed-point properties, the application of an implicit QDEQ model training is superior compared to a set of explicitly repeated layers in almost all cases, with the exception of amplitude-encoded data on our experiment with only four classes. Generally, our experiments indicate that the accuracy of our model is competitive with baseline models that typically require more classical and quantum resources.

One of the key benefits of the DEQ approach is that it avoids the computationally expensive differentiation through the root finder. However, measurements are still required during the forward pass to guide the root finder's Broyden-based updates. The cost of these measurements is likely comparable to the parameter-shift rule commonly used in gradient-based optimization. Further investigation is needed to quantify the measurement overhead and impact on the optimization process. Additionally, the influence of shot noise and other quantum noise sources on DEQ-based training remains to be explored, the expectation being that the impact of noise on DEQ training will behave similarly to general QNN training. Understanding the training of quantum models that are often amenable to vanishing gradients, called barren plateaus, has been greatly advanced by some recent theory works (Fontana et al., 2024; Ragone et al., 2024). We anticipate QDEQ to behave similarly with respect to vanishing gradients as general VQAs. More in-depth analysis is an open problem. The potential advantage of DEQ here is the option of using shallower circuits for a given performance, which are known to be less prone to barren plateaus.

Another avenue of further research could be considering the outcomes of a preceding quantum experiment as quantum data. This could be a physical experiment or a preceding quantum algorithm. Either can be seen as an advanced encoding map. Further investigations towards this approach, and into whether any advantages transfer, is left for further research. Finally, it would be interesting to extend the applicability of DEQs in combination with our deliberations in Observation 2 to a broader range of QML models.

## Acknowledgements

We thank Zachary Cetinic for his insights and rooting for the ultimate side-quest. AAG thanks Anders G. Frøseth for his generous support and acknowledges the generous support of Natural Resources Canada and the Canada 150 Research Chairs program. Parts of the resources used in preparing this research were provided by the Province of Ontario, the Government of Canada through CIFAR, and companies sponsoring the Vector Institute. LBK acknowledges support from the Carlsberg Foundation, grant CF-21-0669. RAVH acknowledges NSERC Discovery Grant No. RGPIN-2024-06594.

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

# Appendix

## A Existence of fixed points for quantum model families

**Observation 2** (Contractiveness of Quantum Model Families).

Below, we provide a discussion regarding the existence of fixed points for quantum model functions according to Definition 1, with the additional assumption that measurement ensembles only consist of Pauli operators and projectors on computational basis states.

Consider two vectors $\mathbf{z}, \mathbf{z}'$, which, without loss of generality, we may assume to be normalized over $[0; 2\pi]^n$. Further, consider a Hermitian observable $M$ used for readout. The value of this readout for quantum models as in Definition 1 then takes the form

$$\mathrm{Tr}\left(M U S_{\mathbf{z}} \left|0\right\rangle\!\left\langle 0\right| S_{\mathbf{z}}^\dagger U^\dagger\right). \tag{6}$$

Now, if we want to evaluate contractiveness of the quantum model and consider

$$\left\| f^{\{M\}}(\mathbf{z}) - f^{\{M\}}(\mathbf{z}') \right\| \tag{7}$$

for a family of quantum models, we can apply the triangle inequality to arrive at conditions for each entry corresponding to a single observable of the ensemble. Thus, to argue whether the model is contractive, we need to bound quantities of the form

$$\Delta = \left| \mathrm{Tr}\left(M U S_{\mathbf{z}} \left|0\right\rangle\!\left\langle 0\right| S_{\mathbf{z}}^\dagger U^\dagger\right) - \mathrm{Tr}\left(M U S_{\mathbf{z}'} \left|0\right\rangle\!\left\langle 0\right| S_{\mathbf{z}'}^\dagger U^\dagger\right) \right|. \tag{8}$$

We use notation as before and say $S_{\mathbf{z}} \left|0\right\rangle = \left|z\right\rangle$. Note that due to the cyclic properties of the trace, we can construct an equivalent observable $\tilde{M} = U^\dagger M U$ with the same spectrum as $M$, and write this as

$$\Delta = \left| \mathrm{Tr}\left( \tilde{M} \left( \left|\mathbf{z}\right\rangle\!\left\langle\mathbf{z}\right| - \left|\mathbf{z}'\right\rangle\!\left\langle\mathbf{z}'\right| \right) \right) \right|. \tag{9}$$

We furthermore can express $\tilde{M}$ in its eigenbasis, $\tilde{M} = \sum_\lambda M_\lambda \left|\lambda\right\rangle\!\left\langle\lambda\right|$. Then evaluating the trace in this basis allows us to simplify the expression as

$$\Delta = \left| \sum_\lambda M_\lambda \left( \left|\left\langle\lambda|\mathbf{z}\right\rangle\right|^2 - \left|\left\langle\lambda|\mathbf{z}'\right\rangle\right|^2 \right) \right|. \tag{10}$$

Now we assume that all eigenvalues of $M$ are bounded, i.e., that it has a finite spectral norm $\|M\| < \infty$. This spectral bound and another application of the triangle inequality yields

$$\Delta \leq \|M\| \sum_\lambda \left| \left|\left\langle\lambda|\mathbf{z}\right\rangle\right|^2 - \left|\left\langle\lambda|\mathbf{z}'\right\rangle\right|^2 \right|, \tag{11}$$

where the right hand side is the total variation distance with respect to some POVM. As such, it is bounded by the trace norm. Then, we can conclude that

$$\Delta \leq 2\|M\| \, \left\| \left|\mathbf{z}\right\rangle\!\left\langle\mathbf{z}\right| - \left|\mathbf{z}'\right\rangle\!\left\langle\mathbf{z}'\right| \right\|_{\mathrm{Tr}} = 2\|M\| \sqrt{1 - \left|\left\langle\mathbf{z}|\mathbf{z}'\right\rangle\right|^2}. \tag{12}$$

As detailed in Appendix C, this bound can be further sharpened to not require the factor of two in the case of basis-state measurements, i.e.,

$$\Delta \leq \|M\| \sqrt{1 - \left|\left\langle\mathbf{z}|\mathbf{z}'\right\rangle\right|^2}. \tag{13}$$

For further discussion of the general case, please see the end of this appendix. Note that in the case of a single observable, this expressions is clearly bounded by $\|M\|$. Assuming that this is smaller than or equal to one, which is clearly a necessary assumption for a contraction map to be possible, we therefore have that the distances after application of the model are always less than one, meaning for any pair of vectors such that $\|\mathbf{z} - \mathbf{z}'\| > 1$, a map of this is always subcontractive. Thus, for the case of a single observable $M$, we only need to consider the case where $\|\mathbf{z} - \mathbf{z}'\| \leq 1$.

Assume that the following holds for some $c > 0$:

$$\left|\left\langle\mathbf{z}|\mathbf{z}'\right\rangle\right| \geq 1 - \frac{c}{2} \sin\left( \|\mathbf{z} - \mathbf{z}'\|^2 \right). \tag{14}$$

Evidence for this bound with $c = 1$ in the case of amplitude encoding can be found in Appendix B, along with evidence for a slightly weaker $c = 2$ bound for the specific angle encoding strategy used in this work. From the inequality, we have the following:

$$1 - |\langle \mathbf{z} | \mathbf{z}' \rangle|^2 \leq 1 - \left(1 - \frac{c}{2} \sin\left(\|\mathbf{z} - \mathbf{z}'\|^2\right)\right)^2 \tag{15}$$

$$= c \sin\left(\|\mathbf{z} - \mathbf{z}'\|^2\right) - \frac{c^4}{4} \sin\left(\|\mathbf{z} - \mathbf{z}'\|^2\right)^2 \tag{16}$$

$$\leq c \sin\left(\|\mathbf{z} - \mathbf{z}'\|^2\right) \leq c \|\mathbf{z} - \mathbf{z}'\|^2 , \tag{17}$$

with the final result that

$$|f(\mathbf{z}) - f(\mathbf{z}')| \leq c \|M\| \|\mathbf{z} - \mathbf{z}'\| \tag{18}$$

for a single output variable so that $\|M\| \leq 1$. Note that the constant $c$ appears as an overall scale in the expression. For simplicity, we will focus on the amplitude encoding case of $c = 1$ below—See Appendix B.2 for a discussion of how the angle-encoding case differs.

Having bounded a single observable, we next discuss how to deal with multiple observables in a family of model functions. If we have an ensemble for $K$ observables, $\{M_k\}_{k=1}^K$, then by the properties of the $\ell_2$ norm,

$$\left\|f^{\{M\}}(\mathbf{z}) - f^{\{M\}}(\mathbf{z}')\right\|_{\ell_2} \leq \sqrt{\sum_k \|M_k\|^2} \|\mathbf{z} - \mathbf{z}'\|_{\ell_2}. \tag{19}$$

So if all $\|M_k\|$ are bounded by 1, we will have a prefactor of $\sqrt{K}$ for $K$ observables, which suggests that for more than one observable, the model family is not (sub)contractive in the $\ell_2$ norm. However, the case would be different if we choose the maximum-norm as a metric. Then, we do not encounter the usual property of the $\ell_2$ norm that it grows with the square-root of the dimension, and we instead have

$$\left\|f^{\{M\}}(\mathbf{z}) - f^{\{M\}}(\mathbf{z}')\right\|_{\max} \leq \max_k \|M_k\| \|\mathbf{z} - \mathbf{z}'\|_{\max}. \tag{20}$$

This means, that our model (family) $f$ from Definition 1 is subcontractive, which is not sufficient yet for a fixed point to exist. One way to proceed would be to show that

$$\|f(f(\mathbf{z})) - f(\mathbf{z})\| < \|f(\mathbf{z}) - \mathbf{z}\| \tag{21}$$

except for $f(\mathbf{z}) = \mathbf{z}$, i.e. unless $\mathbf{z}$ is a fixed point.

Instead, we make make a different deliberation. We restrict ourselves to observables that are either Pauli-operations, which are unitary and have eigenvalues $\pm 1$, or projectors on basis states, whose $+1$ eigenspace is spanned by a single basis state only, as they are rank-one. Note than the bound we used above in Eq. (11) can be loose, as it assumes a worst-case where the difference is aligned in eigenspaces so that $|\mathrm{Tr}(M(|\mathbf{z}\rangle\langle\mathbf{z}| - |\mathbf{z}'\rangle\langle\mathbf{z}'|))| = 2\|M\|\||\mathbf{z}\rangle\langle\mathbf{z}| - |\mathbf{z}'\rangle\langle\mathbf{z}'|\|_{\mathrm{Tr}}$ is attained. Such a scenario is highly unlikely. In fact, consider the relationship in Eq. (21), and denote $|f(\mathbf{z})\rangle = S_{f(\mathbf{z})} |0\rangle$. Then, we look at

$$\|f(f(\mathbf{z})) - f(\mathbf{z})\| = \left| \mathrm{Tr}\left(M(|f(\mathbf{z})\rangle\langle f(\mathbf{z})| - |\mathbf{z}\rangle\langle\mathbf{z}|)\right) \right|. \tag{22}$$

By the arguments of Appendix C, a quantum model is then not contractive if the difference $|f(\mathbf{z})\rangle\langle f(\mathbf{z})| - |\mathbf{z}\rangle\langle\mathbf{z}|$ is fully aligned with either the $+1$ and $-1$ eigenspaces for Pauli operators or the $+1$ eigenspace of a basis state projection, in the sense outlined in that appendix. Surely, as $\mathbf{z}$ comes from a batch of data with different values for each element, such a construction would either not correspond to a sensible set of observables or the encoding would not capture any diversity in the data. Thus, we can expect that the quantum models on data will act as contractions in practice with high probability. This also likely explains why the architectures not fully covered by the analysis above (i.e., angle encoding and Pauli measurements) work in practice.

For the case of amplitude encoding, a re-normalization of the outputs would be required before re-encoding them. Viewing this necessary rescaling of $f^{\{M\}}(\mathbf{z})$ as part of the map may result in a change in norms, since in general $\|\mathbf{x} - \mathbf{y}\| \neq \left\|\frac{\mathbf{x}}{\|\mathbf{x}\|} - \frac{\mathbf{y}}{\|\mathbf{y}\|}\right\|$. Given the positive empirical results of the main text, we conjecture that the arguments for strict contractivity put forth above in most cases more than compensates for any such shifts; we leave further investigation as a topic for future work.

These discussions identify two key aspects to achieve contractive quantum architectures of the form considered in this work: The encoding should encode similar vectors into similar quantum states; and, the measurement operators should not amplify these differences when converting back into classical data. The interplay of these two aspects is captured well by Eq. (9), which highlights the simple fact that the difference in output depends on the difference in the encoded states and the ability of the measurement to resolve this difference. Our approach here used the bound in Eq. (14) as a way to quantify that the encoding preserves closeness, and bounded the resolving/amplification power of the readout using simple operator norms, leaving more detailed analysis of the interplay between encoding and readout as future work.

## B   Evidence for overlap bounds

In Appendix A, the following property for the encoding of vectors fulfilling $\|\mathbf{z} - \mathbf{z}'\| \leq 1$ was used:

$$|\langle \mathbf{z}|\mathbf{z}'\rangle| \geq 1 - \frac{c}{2} \sin\left(\|\mathbf{z} - \mathbf{z}'\|^2\right). \tag{23}$$

In this section, we provide evidence for this bound for the two encoding maps used in the main text, i.e., amplitude encoding and angle encoding.

### B.1   Amplitude encoding

This is the simplest case. Assuming that the inputs are normalized to length one, and using that we only work with real vectors, the following holds by definition of amplitude encoding:

$$|\langle \mathbf{z}|\mathbf{z}'\rangle| = |\mathbf{z} \cdot \mathbf{z}'| \ . \tag{24}$$

Furthermore,

$$\|\mathbf{z} - \mathbf{z}'\|^2 = \|\mathbf{z}\|^2 + \|\mathbf{z}'\|^2 - 2\mathbf{z} \cdot \mathbf{z}' \tag{25}$$
$$= 2 - 2\mathbf{z} \cdot \mathbf{z}' , \tag{26}$$

meaning

$$|\langle \mathbf{z}|\mathbf{z}'\rangle| = \left|1 - \frac{1}{2}\|\mathbf{z} - \mathbf{z}'\|^2\right|. \tag{27}$$

Using the fact that the norm in this expression is bounded by one, we therefore get

$$|\langle \mathbf{z}|\mathbf{z}'\rangle| = 1 - \frac{1}{2}\|\mathbf{z} - \mathbf{z}'\|^2 \tag{28}$$
$$\geq 1 - \frac{1}{2}\sin\left(\|\mathbf{z} - \mathbf{z}'\|^2\right), \tag{29}$$

as desired. Note that for this particular encoding, the equality in Eq. (28) allows for the introduction of a factor $\frac{1}{2}$ in Eq. (17), corresponding to the the constant $c = 1$ in the bound in Eq. (23).

### B.2   Angle encoding

In the case of the angle encoding, consider the unitaries used in the encoding:

$$|\langle \mathbf{z}|\mathbf{z}'\rangle| = \left|\langle 0| S_{\mathbf{z}}^{\dagger} S_{\mathbf{z}'} |0\rangle\right|. \tag{30}$$

For the specific encoding used, these unitaries have the property that they consist of single-qubit encodings operating on separate qubits. In other words, denoting these single-qubit maps by $S_{\mathbf{z}}^{(1)}$ and splitting the vectors into their constituent 4-tuples of entries,

$$\mathbf{z}_k = \left[(\mathbf{z})_{4k+1}, (\mathbf{z})_{4k+2}, (\mathbf{z})_{4k+3}, (\mathbf{z})_{4k+4}\right]^T, \tag{31}$$

we can write the encoding map as the following tensor product

$$S_{\mathbf{z}} = \bigotimes_{k=1}^{Q} S_{\mathbf{z}_k}^{(1)}. \tag{32}$$

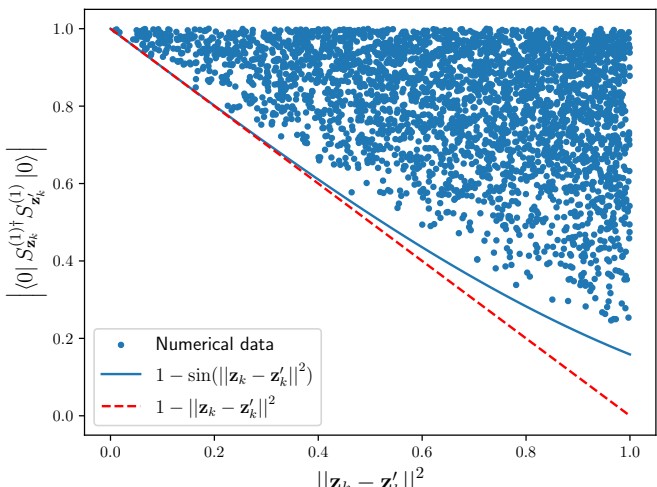

Figure 4: Plots of the relation between the single-qubit overlaps $\left| \langle 0 | S_{\mathbf{z}_k}^{(1)\dagger} S_{\mathbf{z}_k'}^{(1)} | 0 \rangle \right|$ and the norm $\| \mathbf{z}_k - \mathbf{z}_k' \|^2$ for 3000 pairs of random vectors in $\mathbb{R}^4$. Note that all of the points lie above the line corresponding to $1 - \sin\left( \| \mathbf{z}_k - \mathbf{z}_k' \|^2 \right)$.

This implies that the overlap takes the form:

$$| \langle \mathbf{z} | \mathbf{z}' \rangle | = \prod_{k=1}^{Q} \left| \langle 0 | S_{\mathbf{z}_k}^{(1)\dagger} S_{\mathbf{z}_k'}^{(1)} | 0 \rangle \right|. \tag{33}$$

Each factor in this expression is in principle simply a complicated trigonometric expression in eight variables. While it is likely possible to bound this expression analytically, a simple numerical investigation was assumed sufficient evidence for our investigation here; see Fig. 4. Drawing 3000 pairs of vectors $\mathbf{z}_k, \mathbf{z}_k' \in \mathbb{R}^4$ at random so that $\| \mathbf{z}_k - \mathbf{z}_k' \| \leq 1$, we see that in all cases the following inequality holds,

$$\left| \langle 0 | S_{\mathbf{z}_k}^{(1)\dagger} S_{\mathbf{z}_k'}^{(1)} | 0 \rangle \right| \geq 1 - \sin\left( \| \mathbf{z}_k - \mathbf{z}_k' \|^2 \right). \tag{34}$$

To finish the bound, we apply the following trigonometric identity,

$$\sin(a + b) = \sin(a)\cos(b) + \sin(b)\cos(a) \tag{35}$$

and use this to derive:

$$\begin{align}
(1 - \sin(a))(1 - \sin(b)) &= 1 - \sin(a) - \sin(b) - 2\sin(a)\sin(b) \tag{36} \\
&= 1 - \sin(a) - \sin(b) - 2\sin(a)\sin(b) \tag{37} \\
&\quad + \sin(a)\cos(b) + \sin(b)\cos(a) - \sin(a+b) \tag{38} \\
&= 1 - \sin(a+b) \tag{39} \\
&\quad + \sin(a)(\cos(b) + \sin(b) - 1) \tag{40} \\
&\quad + \sin(b)(\cos(a) + \sin(a) - 1). \tag{41}
\end{align}$$

For $0 < a, b < 1$, the two final terms are either positive or zero. Thus, under this assumption on the arguments, we obtain

$$(1 - \sin(a))(1 - \sin(b)) \geq 1 - \sin(a+b). \tag{42}$$

Combining Eq. (33) with the bounds of Eq. (34) and Eq. (42) now yields

$$|\langle \mathbf{z}|\mathbf{z}'\rangle| = \prod_{k=1}^{Q} \left| \langle 0| S_{\mathbf{z}_k}^{(1)\dagger} S_{\mathbf{z}_k'}^{(1)} |0\rangle \right| \tag{43}$$

$$\geq \prod_{k=1}^{Q} \left( 1 - \sin\left( \|\mathbf{z}_k - \mathbf{z}_k'\|^2 \right) \right) \tag{44}$$

$$\geq 1 - \sin\left( \sum_k \|\mathbf{z}_k - \mathbf{z}_k'\|^2 \right) \tag{45}$$

$$= 1 - \sin\left( \|\mathbf{z} - \mathbf{z}'\|^2 \right), \tag{46}$$

as desired.

Note that, in contrast to the amplitude-encoding case, we do not have a pre-factor of $1/2$ on the second term. In other words, our bound corresponds to the one in Eq. (23) with $c = 2$. As can be seen by the derivations in Appendix A, this introduces an overall factor of 2 that would in principle need to be compensated for, e.g., in the magnitude of the readout operators. However, our experiments indicate no problems finding fixed points also without such adaptations. We ascribe this success to the likely looseness of the bounds derived in Appendix A, since this means contractiveness can be present even when the derived bounds are not sufficient to guarantee it. For a further discussion of some of the sources of this looseness, see Appendix A.

## C  Further analysis of bound tightness

In this section, we provide a more detailed analysis of the steps leading from Eq. (9) to Eq. (12) in Appendix A, including the derivation of a tighter bound for the case of observables that are projectors.

Consider the object to be bounded,

$$\Delta = \left| \text{Tr}\left( \tilde{M} \left( |\mathbf{z}\rangle\langle\mathbf{z}| - |\mathbf{z}'\rangle\langle\mathbf{z}'| \right) \right) \right|. \tag{47}$$

Looking more closely at the object $|\mathbf{z}\rangle\langle\mathbf{z}| - |\mathbf{z}'\rangle\langle\mathbf{z}'|$, we can note that this is a Hermitian, traceless rank-2 operator. This implies that it can be diagonalized unitarily, with at most two nonzero eigenvalues that necessarily sum to zero due to the tracelessness. In other words, it can be written on the form

$$|\mathbf{z}\rangle\langle\mathbf{z}| - |\mathbf{z}'\rangle\langle\mathbf{z}'| = \lambda \left( |\xi_0\rangle\langle\xi_0| - |\xi_1\rangle\langle\xi_1| \right) \tag{48}$$

for some $\lambda \geq 0$. In fact, an explicit calculation shows that this constant can be characterized as $\lambda = \||\mathbf{z}\rangle\langle\mathbf{z}| - |\mathbf{z}'\rangle\langle\mathbf{z}'|\|_{\text{Tr}}$. Thus, Eq. (47) can be rewritten as

$$\Delta = \left| \text{Tr}\left( \tilde{M} \left( |\xi_0\rangle\langle\xi_0| - |\xi_1\rangle\langle\xi_1| \right) \right) \right| \||\mathbf{z}\rangle\langle\mathbf{z}| - |\mathbf{z}'\rangle\langle\mathbf{z}'|\|_{\text{Tr}} \tag{49}$$

$$= \left| \text{Tr}\left( \tilde{M} |\xi_0\rangle\langle\xi_0| \right) - \text{Tr}\left( \tilde{M} |\xi_1\rangle\langle\xi_1| \right) \right| \||\mathbf{z}\rangle\langle\mathbf{z}| - |\mathbf{z}'\rangle\langle\mathbf{z}'|\|_{\text{Tr}}. \tag{50}$$

Comparing this to the bound in Eq. (12),

$$\Delta \leq 2\|M\| \||\mathbf{z}\rangle\langle\mathbf{z}| - |\mathbf{z}'\rangle\langle\mathbf{z}'|\|_{\text{Tr}}, \tag{51}$$

it becomes clear that the tightness of the bound depends on how effectively the observable $\tilde{M}$ distinguishes between the two eigenstates of $|\mathbf{z}\rangle\langle\mathbf{z}| - |\mathbf{z}'\rangle\langle\mathbf{z}'|$. This, in turn, depends on the alignment of these eigenstates with the eigenspaces of the measurement operator $\tilde{M}$. Specifically, in the case where the spectrum of $M$ takes the form $\{-\|M\|, \ldots, \|M\|\}$, the bound is saturated when one of the eigenstates is contained in the $+\|M\|$-eigenspace of $\tilde{M}$ and the other one is contained in the $-\|M\|$-eigenspace of $\tilde{M}$. On the other hand, for operators with spectra of the form $\{0, \ldots, \|M\|\}$ (e.g., basis-state projectors), the expression in Eq. (49) is maximized when one eigenstate is fully contained in the 0-eigenspace and one is fully contained in the $+\|M\|$-eigenspace. Note that, in this case, the following tighter bound holds:

$$\Delta \leq \|M\| \||\mathbf{z}\rangle\langle\mathbf{z}| - |\mathbf{z}'\rangle\langle\mathbf{z}'|\|_{\text{Tr}}, \tag{52}$$

as used in Appendix A.

## D  Universality of weight-tied quantum models

**Theorem 3** (Universality of weight-tied quantum models, in analogy to Theorem 3 in Bai et al. (2019))**.**
*Let $\mathcal{E}_i(\cdot) = U(\theta^{(i)})(\cdot)U^\dagger(\theta^{(i)})$ be a channel corresponding to the PQC at depth $i$. Additionally, we define an map $R' : \mathbb{C}^{2^Q \times 2^Q} \to \mathbb{R}^K$ that describes performing measurements of expectation values with respect to an ensemble of $K$ observables and storing the respective outcomes in a $K$-dimensional vector. Typically, we string this together with an upsampling map $\mathcal{I}_u$ so that $R = \mathcal{I}_u \circ R'$ maps to $\mathbb{R}^n$. Let $S_{\mathbf{z}}$ be a unitary encoding that encodes a vector from $\mathbb{R}^n$ into a quantum state in $\mathbb{C}^{2^Q}$, $S_{\mathbf{z}} : |0\rangle \mapsto |\mathbf{z}\rangle$. This allows us to define an encoding map $\mathcal{S}$ that maps $\mathbf{z}$ to a density matrix, so that $\mathbf{z} \mapsto |\mathbf{z}\rangle\langle\mathbf{z}|$, where the evolution of hidden layers can then be written as*

$$\mathbf{z}^{(i+1)} = R(\mathcal{E}_i \circ \mathcal{S}\mathbf{z}^{(i)}), \quad 0 \leq i < L, \quad \mathbf{z}^{(0)} = \mathbf{x}. \tag{53}$$

*Then, a sequence of such layers can be replicated exactly by an input-injected, weight-tied network. Specifically,*

$$\widetilde{\mathbf{z}}^{(i+1)} = R_L(E_z \widetilde{\mathbf{z}}^{(i)} + E_x \mathbf{x}) \tag{54}$$

*where*

$$R_L = \begin{bmatrix} R \\ \vdots \\ R \end{bmatrix}, \quad E_z = \begin{bmatrix} 0 & 0 & 0 & \cdots & 0 \\ \mathcal{E}_1 \circ \mathcal{S} & 0 & 0 & \cdots & 0 \\ 0 & \mathcal{E}_2 \circ \mathcal{S} & 0 & \cdots & 0 \\ & & \ddots & & 0 \\ 0 & 0 & 0 & \mathcal{E}_{L-1} \circ \mathcal{S} & 0 \end{bmatrix}, \quad E_x = \begin{bmatrix} \mathcal{E}_0 \circ \mathcal{S} \\ 0 \\ \vdots \\ 0 \end{bmatrix}, \tag{55}$$

*yields after $L$ iterations an output*

$$\widetilde{\mathbf{z}}^{(L)} = \begin{bmatrix} \mathbf{x} \\ \mathbf{z}^{(1)} \\ \vdots \\ \mathbf{z}^{(L)} \end{bmatrix} \tag{56}$$

*containing the output $\mathbf{z}^{(L)}$ of the non weight-tied network. This follows by construction, similarly to Theorem 3 in Bai et al. (2019).*

# E   Experiments – Hyperparameter search

We used the validation set to search over hyperparameters listed in Table 5. For each hyperparameter, we note the ranges we considered, as well as the optimal value for each dataset.

Table 5: Optimal hyperparameters for each model. The search space we considered was: learning rate $\{0.005, 0.0075, 0.01, 0.05, 0.1\}$, number of warm-up steps $\{1875, 2355, 3750\}$, number of warm-up layers $\{1, 2\}$, weight of the Jacobian loss $\{0, 0.5, 0.8\}$, frequency of the Jacobian loss $\{0.0, 0.5, 0.8, 1.0\}$.

| | IMPLICIT+WARMUP | | | |
| Hyperparamter | MNIST-4 | MNIST-10 | FashionMNIST-10 | CIFAR-10 |
| --- | --- | --- | --- | --- |
| Learning rate | 0.05 | 0.05 | 0.05 | 0.01 |
| Num. warm-up steps | 1875 | 1875 | 1875 | 2350 |
| Warm-up layer depth | 1 | 1 | 1 | 1 |
| Jac. loss weight | 0.0 | 0.8 | 0.8 | 0.8 |
| Jac. loss freq. | 0.0 | 1.0 | 0.8 | 1 |

| | IMPLICIT | | | |
| Hyperparamter | MNIST-4 | MNIST-10 | FashionMNIST-10 | CIFAR-10 |
| --- | --- | --- | --- | --- |
| Learning rate | 0.05 | 0.05 | 0.05 | 0.05 |
| Jac. loss weight | 0.0 | 0.8 | 0.5 | 0.0 |
| Jac. loss freq. | 0.0 | 1.0 | 0.8 | 0.0 |

| | DIRECT - 10 layers | | | |
| Hyperparamter | MNIST-4 | MNIST-10 | FashionMNIST-10 | CIFAR-10 |
| --- | --- | --- | --- | --- |
| Learning rate | 0.1 | 0.05 | 0.05 | 0.0075 |

| | DIRECT - 5 layers | | | |
| Hyperparamter | MNIST-4 | MNIST-10 | FashionMNIST-10 | CIFAR-10 |
| --- | --- | --- | --- | --- |
| Learning rate | 0.05 | 0.05 | 0.0075 | 0.0075 |

| | DIRECT - 2 layers | | | |
| Hyperparamter | MNIST-4 | MNIST-10 | FashionMNIST-10 | CIFAR-10 |
| --- | --- | --- | --- | --- |
| Learning rate | 0.05 | 0.05 | 0.05 | 0.01 |

| | DIRECT - 1 layer | | | |
| Hyperparamter | MNIST-4 | MNIST-10 | FashionMNIST-10 | CIFAR-10 |
| --- | --- | --- | --- | --- |
| Learning rate | 0.05 | 0.05 | 0.01 | 0.01 |

