# OpenReview forum: "Quantum Deep Equilibrium Models"
_NeurIPS.cc/2024/Conference — NeurIPS 2024 poster_

### Official Review · Reviewer_fCtD · 2024-07-07

**Soundness:** 4
**Presentation:** 4
**Contribution:** 3
**Rating:** 6
**Confidence:** 3

**Summary:**

The technique of deep equilibrium models, which were introduced to efficiently handle classical sequential data, is here applied to networks consisting of quantum circuits. The performance is compared to both direct solvers and baseline algorithms (VAE and PCA) for datasets derived from MNIST-4, MNIST, and FashionMNIST. The proposed scheme consistently beats the direct solvers but only sometimes the baselines.

**Strengths:**

Bringing deep equilibrium models to quantum circuits seems promising and this is a significant first step demonstrating a speedup for a concrete example. The article clearly describes the setup and what experiments were done. Also limitations are clearly stated as benchmarking is also performed to reference algorithms and not only more basic architectures.

**Weaknesses:**

All datasets are classical data. It is not clear why one should use quantum circuits for such data except as a first demonstration of the scheme.

**Questions:**

Why were all datasets classical and what advantage might a quantum circuit have in their analysis?
Could quantum datasets be tried instead, and would you expect an advantage there?

**Limitations:**

The limitations are clearly stated in the article.

---

> ### Author Rebuttal · Authors · 2024-08-07
>
> We are grateful for the reviewer’s positive assessment and thoughtful comments on our work. In response to their questions, we offer the following answers.
>
> **Classical vs Quantum Data**
>
> Firstly, we would like to state that we are not intending this work to make any strong statements about quantum advantages for classical data, and we would agree with the reviewer that utility of quantum models for classical data is still an open question. On one hand, applied hybrid quantum/classical methods are receiving a lot of positive attention, and there are supporting theoretical results showing that provable learning advantages on classical data is possible (under certain complexity-theoretic assumptions) on highly artificial problems related to cryptography or contextuality. On the other hand, some claims of advantage have proven hard to reproduce, and there is well-founded scepticism about whether the structure and bias of quantum models aligns with the needs of real-world datasets. Given the differing viewpoints and conflicting evidence, we believe the question open enough that contributing methods to QML for classical data is still meaningful.
>
>
> The application of QDEQ to quantum data would be a very interesting avenue of further research. However, generalizing the framework to quantum-state inputs/outputs and contractive (i.e., dissipative) parametrized quantum channels presents several subtle questions. For instance, how can we efficiently measure state-closeness when doing root-finding? What is the best way allow the root finder to move in some (efficiently parametrizable) subset of quantum states? How can we adapt the training algorithm to take advantage of this reduced space rather than objects scaling with the full $2^{2Q}$ size of the input/output space?
>
> Another promising avenue could be considering the outcomes of a preceding quantum experiment as quantum data. This could be an experiment in the sense of a physical experiment, or also in the sense of a preceding quantum algorithm. Either can be seen as an advanced encoding map. For the case of it coming from an algorithm, the final result aka the ML input data if often delivered in amplitude or phase. While this data in amplitude or phase can be transitioned to and from each other by amplitude amplification and quantum phase estimation, such tools may require fault-tolerance and more advanced quantum hardware. In the case of experimental results, in general, we expect this data to be more ``messy'' and it would require a case-by-case basis how to deal with this specifically. Nevertheless, we expect that in the case of data coming from quantum computations and experiments, QDEQ would fare similarly to e.g. variational quantum circuit alternatives on this analysis task, and also similarly inhibit its strengths and weaknesses compared to standard variational circuit approaches.
>
>
> While the open questions prevented a direct application of QDEQ as presented here to fully quantum data, further research into answering them, and into whether any advantages transfer, does seems promising. Moreover, we are also hopeful that the broader idea of fixed-point search of quantum maps could be a promising algorithmic building block also more generally, i.e., beyond classification tasks. However, due to the subtleties, it is hard to speculate if the advantages of better shallow-circuit performance transfer, and the considerations in general had to be postponed to future work. Some of the above considerations are in some form mentioned very briefly on lines 173-178 of our submission, to be clarified in future revisions.
>
> We sincerely appreciate the reviewer’s constructive feedback and encouraging review. We hope these additional insights clarify any outstanding questions, and we are available to provide more details if necessary.

---

> > ### Comment · Reviewer_fCtD · 2024-08-09
> >
> > Thank you for your response and the perspective on the different types of datasets.

---

> > > ### Author Response · Authors · 2024-08-13
> > >
> > > Thank you for taking the time to review our responses!

---

### Official Review · Reviewer_1Wfk · 2024-07-12

**Soundness:** 2
**Presentation:** 2
**Contribution:** 2
**Rating:** 4
**Confidence:** 4

**Summary:**

In this paper the author present a quantum version of the Deep Equilibrium models. These networks approximate iterative approach through many layers by approximating it with a single set of parameters that would have converged if there were an infinity of layers. The paper is written well and the method is verified on a set of small benchmarks.

**Strengths:**

- The paper presents a quantum generalization of the Deep Equilibrium models for the classification task and implemented through an ensemble methods using the VQA for training.

**Weaknesses:**

- The main disadvantage of the proposed model is the relatively weak result analysis. When looking at the provided results one can conclude that the proposed approach provides in average over one percent accuracy improvement while requiring the double processing time and double memory size. This is when comparing the Implicit solver vs. Single Layer direct learning. Naturally when more layers would be present this could change but with larger dataset the runtime of the implicit solver could change???

Simply based on these observations I wonder about the usefulness. This is because the MNIST dataset is by default the simplest one and is used as a principle demonstrator. So if such a large overhead is present how will it scale for both the accuracy as well as for the requirements (memory and time)?

**Questions:**

It would be interesting to see if and how this approach scales with real datasets such as for instance Cifar.

**Limitations:**

Scalability and novelty.

---

> ### Author Rebuttal · Authors · 2024-08-07
>
> We appreciate the reviewer’s feedback and constructive critique of our work. In response to their concerns, we have provided the following clarifications and additional information:
>
> **Scalability**
>
> We agree with the reviewer that scaling is an important question, and one that is very difficult to meaningfully assess using the small-scale experiments available in classical simulations of quantum models. This is also why the MNIST dataset was chosen, since the small size of simulatable circuits makes full 10-class MNIST meaningfully challenging task for quantum models built without significant classical components for assistance, i.e., models we develop in this work and our baselines. In general, MNIST-4 and MNIST-10 are used frequently in quantum ML settings because quantum simulations are expensive. With respect to the observed scaling, it should be noted that the compute-cost of classical simulation reported here is only an imperfect proxy for the effort required to train the models with access to a quantum computer. In the long-term quantum context, the amount of measurements required to evaluate gradients would be a central measure of complexity, since measurement and reset is often slower than logic gates. However, even more important currently is circuit depth, which constitutes a hard limit to the size of implementable models. In this respect, we believe QDEQ to be worthwhile for the model-sizes considered here. As for the measurement requirements, the relative overhead of QDEQ scales with the size of the input space relative to the size of the parameter space. Currently, baselines have these spaces at roughly the same size. However, the impact of scaling models and/or problem sizes in the future is hard to predict -- one could hope for deep-learning-like scaling where parameter-size$\\gg$data-size, but could also end up being limited by poor optimization landscapes (e.g., the barren plateaus discussed in the response to **BzP4**) to more similar sizes. In the first scenario, the measurement overhead of QDEQ likely becomes insignificant compared to the general effort of estimating gradients wrt. trainable parameters, while the second situation could see more non-negligible overheads.
>
> In other words, for current devices, circuits are still shallow enough that readout constitutes a significant portion of the runtime. However, since noise sets a hard upper limit on the size of implementable models, QDEQ can still bring advantages despite possessing some measurment overhead (related to measuring gradients in data-space). On the other hand, fault tolerant quantum computers may bring much deeper architectures, where the circuit depth dominates over measurement-time, and where the number of trainable parameters is much larger than the size of the data-space. In such a scenario, the measurement overhead of QDEQ (related to the gradients of the now relatively small data-space) has the potential to be insignificantly small compared to the runtime-cost of running twice as deep a circuit. While predicting how quantum models will scale in the future is very difficult, and certain obstacles in trainability would have to be overcome, we believe that the applicability of QDEQ to both near-term and such future scenarios makes it a worthy topic of study, even with the overhead in measurement complexity.
>
> **CIFAR dataset**
>
> At the reviewer's request, we have conducted experiments on CIFAR-10 [1], which consists of real-world images. We show the results in a table below. For CIFAR-10, we first converted the images to greyscale and resized from 32x32 to 28x28 to adapt them to our setup.
>
> | Model    | Test accuracy (\%) |
> | -------- | ------- |
> | Implicit solver [ours]  |  $0.2438$  |
> | Implicit solver + Warmup [ours] |  $\\mathbf{0.2545}$      |
> |  Direct solver [10 layers]  | $0.2371$    |
> |  Direct solver [5 layers]  | $0.2419$     |
> |  Direct solver [2 layers]  | $0.2490$     |
> |  Direct solver [1 layers]  | $0.2470$     |
>
> We find that in general, near-term quantum ML models that are amenable to numerical experiments do not perform well on CIFAR-10, which has also been observed in recent prior works [1] (models do better if a lot of classical NN components are added, but we refrained from doing this). Despite this, we find that our QDEQ framework (Implicit + Warmup) still does better on the test set than the direct solver baseline. This can motivate the utility of our method on more realistic datasets.
>
> [1] Baek H. et al., Logarithmic dimension reduction for quantum neural networks. In Proceedings of the 32nd ACM International Conference on Information and Knowledge Management, 2023.
>
>
> **Novelty**
>
> We believe our work is indeed novel as we develop a new method enabling the first demonstration of applying deep equilibrium models in a quantum machine learning setting.
>
> We are grateful for the reviewer’s evaluation of our study. We hope our responses address the raised concerns and provide further clarity. We are available to offer additional details if needed.

---

### Official Review · Reviewer_BzP4 · 2024-07-13

**Soundness:** 3
**Presentation:** 3
**Contribution:** 3
**Rating:** 6
**Confidence:** 2

**Summary:**

This paper introduces a new paradigm for training quantum machine learning models using Deep Equilibrium Models (DEQ). The authors propose Quantum Deep Equilibrium Models (QDEQ) to enhance the performance of parametrized quantum circuits (PQC) while addressing issues related to circuit depth and parameter scalability. QDEQ uses a root solver to find the fixed points of the network, allowing for the training of shallower circuits with fewer parameters. The paper demonstrates the effectiveness of QDEQ on classification tasks using MNIST-4, MNIST-10, and FashionMNIST-10 datasets, showing competitive performance compared to existing baseline models while requiring fewer resources.

**Strengths:**

The paper introduces the Deep Equilibrium Models to quantum machine learning, a novel approach that has not been explored previously.

Using DEQ, the paper shows that it is possible to achieve high performance with significantly shallower quantum circuits, which is crucial for near-term quantum devices where noise is a significant factor.

The paper provides a detailed and solid theoretical framework.

The presentation of the paper is very easy to understand, with a good mix of equations/theorems and explanations/illustrations.

The paper demonstrates experiments on MNIST-4, MNIST-10, and FashionMNIST-10 datasets, showing the practical applicability and competitive performance of QDEQ models, and adding empirical evidence to the claim.

The proposed approach is resource efficient and reduces the need for extensive quantum and classical resources, making it more feasible for implementation on current quantum hardware.

**Weaknesses:**

The effectiveness of QDEQ relies on the assumption that the quantum model functions admit fixed points. It is unclear how well this assumption can hold in reality

The empirical validation is limited to a few datasets. The reviewer understands the complexity of performing experiments for quantum machine learning, so this point can be ignored.

There is only one or two baselines in each benchmark. Additional baselines could help to further validate the result.

Although the authors did a lot of work to make the work reproducible, the code is not publicly available yet. The authors promised to post the code after the review, but in the meantime, it would be helpful to have an anonymous repository to host the code.

**Questions:**

How does quantum noise, such as shot noise and other errors, influence the DEQ-based training process on noisy quantum hardware?

In practical implementations, how often do the assumptions about the existence of fixed points hold, and what are the implications if they do not?

Can the authors comment on how this work is related to the know barrel plateau issue in quantum machine learning?

**Limitations:**

The limitation and broader impact (which may not exist) are addressed.

---

> ### Author Rebuttal · Authors · 2024-08-07
>
> We appreciate the reviewer’s insightful comments and are thankful for the positive feedback. In response to the reviewer’s questions, we have provided the following answers:
>
> **Influence of noise**
>
> We agree that this is a very interesting and relevant question that we opted to defer to further research -- the emphasis being, noise is a very wide and subtle topic and we would like to spend more time on this question in a thorough manner in a follow-up work. Furthermore, accurately modelling it requires code modifications beyond what was feasible in a week of rebuttal.
>
> The main noise-sensitive component of QDEQ is that gradients need to be retrieved through sampling, which makes the training potentially sensitive to both shot noise and other noise sources. In this sense, we do not expect QDEQ to be significantly different to standard variational algorithms, with similar error mitigation strategies likely being applicable here.
>
> Note that some noise sources, e.g., amplitude damping noise, might aid the existence of fixed points. However, in the presence of strong noise, these fixed points tend to be trivial (maximally mixed state) and thus likely less useful. As such, a closer investigation is required on whether the impact of intermediate noise levels would be beneficial.
>
> **Fixed points in practice**
>
> We acknowledge that the question of fixed points is important and not fully closed by our work. We consider the presence of fixed points in the architectures we investigated as promising empirical evidence. Contractions are guaranteed to have fixed points, thus Appendix A of our manuscript aims to provide guidelines regarding which properties are important for ensuring this property. The two most important factors are the encoding and the readout. Encoding should not amplify differences between two data points, i.e., neighbouring data points should encode to similar quantum states. The amplitude encoding we use satisfies this if the classical vector to be encoded is not very small. This is supported by input injection in our approach, so this should be satisfied for nontrivial datasets. Other widely-used encodings, such as angle encoding, have the same property. With respect to readout, similar non-amplification is needed. This implies that the norm of the measured operator should not be too large and that there is an upper bound on how well it distinguishes encoded states, in the sense of how states are aligned with the measurement operator's eigenbasis , cf. Eq. (9) or (12). The norm is easy to calculate and relatively small for single-qubit Paulis and basis-state projections, as common in quantum classifiers. The distinguishing-power is more difficult to bound analytically. Based on this and the empirics, we expect fixed points to be common for quantum classifier architectures. Further work on a formal proof, e.g. using arXiv:2006.08591, is left as future work.
>
> If the map completely lacks fixed points, we expect training of the model to fail; the essential implicit differentiation formalism breaks down. We briefly investigated this by trying to fit our model to a Fourier Series --  there is no fixed point in this setting and thus QDEQ performed well below the baseline model.
>
> **Benchmarks**
>
> We agree that baselines are vital to establish a context for the presented results. We expanded our search for admissible baselines and have added a new baseline by arXiv:2309.09424 to the MNIST-10 and FashionMNIST-10 benchmarks. This reference provides evidence of performance of $\\sim$80-85\\% on MNIST-10 and $\\sim$75\\% on FashionMNIST-10 using a slightly modified amplitude encoding and a similar number of qubits. Additionally, the baseline we already included in the submission for FashionMNIST-10 contains more experiments in its appendix. These experiments show performance of $77-82\\%$, and use non-approximate amplitude encoding, similar to our strategy. It should be noted that in both cases, the high performance is likely attributable to significantly deeper circuits with more trainable parameters (about 10x and 3x the parameters of our architecture, respectively). We see no limitations to scaling QDEQ to similar sizes, and leave this as future work.
>
> Beyond these references, existing work is dominated by binary classification, convolutional architectures, or significant initial compression of the images using deep classical NNs. We are concerned that adding baselines with very different architectures or tasks would not support the main focus of the paper -- comparing QDEQ to other training of similar architectures -- and that extended discussions of architectural differences risks reducing the clarity of the paper. As we see it, comparisons between the direct solver and the implicit solver should be the focal point of the discussion, as this most cleanly reflects the effects of QDEQ.  We hope that the reviewer will find these additions an reasonable trade-off between context and focus.
>
> **Barren plateaus**
>
> The training (and evaluation) of the models still relies on gradients evaluated by measurement. As such, we would expect the barren plateau issues of the model function to be identical to those of other parametrized quantum circuits, presenting similar challenges and being amenable to several known mitigation strategies. One such  mitigation approach, local observables, is present in our experiments. We expect the need to evaluate gradients with respect to $z$ to behave similarly, especially in the case of angle encoding. As we see it, the main bearing of QDEQ on barren plateau issues would be as a potential way of pushing the performance of shallow trainable models further, to some extend mitigating the need to increase circuit depth to hard-to-train levels.
>
> Once again, we thank the reviewer for their thoughtful feedback and positive evaluation of our work. We hope these additional comments address any remaining questions and are happy to provide further details if needed.

---

> > ### Comment · Reviewer_BzP4 · 2024-08-11
> >
> > Thanks a lot for taking the time to answer my questions! I would like to keep my recommendation for acceptance of the paper.

---

> > > ### Author Response · Authors · 2024-08-13
> > >
> > > Thanks very much for your positive feedback!

---

> ### Author Response · Authors · 2024-08-08
> **Anonymized repository**
>
> Dear reviewer, the AC gave permission to share this link to the repository now:
>
> https://anonymous.4open.science/r/qdeq-neurips-F57D/
>
> Best,
> Authors

---

### Author Rebuttal · Authors · 2024-08-07

We  sincerely  thank the reviewers for spending time on our paper and providing valuable feedback. We are glad that the reviewers find our generalization of deep equilibrium networks to quantum circuits novel and interesting for real-world applications.

We have addressed the following points in response to the reviewers comments:
1. Performed experiments on an additional dataset, CIFAR-10. As requested by Reviewer **1Wfk**, we demonstrated our approach on CIFAR10 -- a more realistic dataset. We note that CIFAR10 is a difficult benchmark for current quantum frameworks in general, but we show that using our deep equilibrium framework we can achieve higher performance than with direct solvers, and comparably with fully-quantum baselines. This brings the number of datasets where QDEQ is demonstrated to improve performance to three.
2.   Discussed the impact of (shot) noise.
3. Provided additional argumentation with respect to the  barren plateau issue and the applicability of QDEQ to quantum data.
4.  Increased the number of baselines we compare our work to.
5.  Provided a link to an anonymous repository with our codebase. As per the NeurIPS rebuttal policy, we have shared the link with the area chair to forward to the reviewers.

We would like to bring to the reviewers attention that we found a mistake introducing a factor of two in Eq. (17) in the appendix of our submission. This does not change the contractiveness results for the Amplitude encoding -- the main encoding used in this work -- due to cancelling with a factor of $\\frac{1}{2}$ from Eq. (28). However, it does weaken the result for the specific angle encoding used in the MNIST-4 experiments. Since the appendix is only intended to provide observations to elucidate and support the main evidence -- the empirical tests -- this does not change the conclusions of the paper. Nevertheless, we wanted to bring it to the reviewers' attention for the sake of transparency. For more details, see also the discussion of contractiveness in the response to reviewer **1Wfk** below.

---

### Decision · Program_Chairs · 2024-09-25

**Decision:**

Accept (poster)

**Comment:**

The paper extends the deep equilibrium model (DEQ) formulation to quantum machine learning models, and demonstrates that the proposed QDEQ models can achieve better accuracy given same amount of model parameters. Reviewers generally appreciated the novelty and soundness, while also raised concerns including analysis on the influence of noise, insufficient improvement using shallow models, scalability challenge, and limited evaluation datasets. On these points, in the rebuttal the authors made an explanation on the conditions for the method to work, the restriction and potential to carry out larger-scale simulation, and presented additional results on the CIFAR-10 dataset which is relatively larger, and explained the applicability to quantum data. The response seems to have addressed most of the concerns, while others seem reasonably responded. I therefore recommend an accept to the submission.